# Adaptation of the PERCEPT myeloma prehabilitation trial to virtual delivery: changes in response to the COVID-19 pandemic

Orla McCourt ,[1,2] Abigail Fisher,[3] Gita Ramdharry,[4] Anna L Roberts,[3] Joanne Land,[3] Neil Rabin,[5] Katie Rowe,[2] Kwee Yong[2]

[1]Therapies & Rehabilitation, University College London Hospitals NHS Foundation Trust, London, UK
[2]Research Department of Haematology, Cancer Institute, University College London, London, UK
[3]Research Department of Behavoural Science and Health, University College London, London, UK
[4]Queen Square Centre for Neuromuscular Diseases, National Hosital for Neurology and Neurosurger, London, UK
[5]Department of Haematology, University College London Hospitals NHS Foundation Trust, London, UK

**Correspondence to**
Ms Orla McCourt;
o.mccourt@nhs.net

## ABSTRACT

**Introduction and objective** Research activity was impacted by the novel COVID-19 pandemic, the PERCEPT myeloma trial was no exception. This pilot randomised trial delivered a face-to-face exercise intervention prior to and during autologous stem cell transplantation (ASCT) in myeloma patients, as a consequence of COVID-19 it required significant adaptions to continue. This brief communication describes how the previously published study protocol was adapted for virtual delivery. In addition, we highlight the challenge of continuing the study which was embedded within a clinical pathway also impacted by the pandemic.

**Summary** The original trial protocol was amended and continued to recruit and deliver an exercise prehabilitation intervention virtually. Continued delivery of the intervention was deemed important to participants already enrolled within the trial and the adapted virtual version of the trial was acceptable to the research ethics committee as well as participants. Development of effective, remotely delivered rehabilitation and physical activity programmes are likely to benefit people living with myeloma. The COVID-19 pandemic provided an opportunity to explore the feasibility of a virtual programme for ASCT recipients, however, continued changes to the clinical pathway within which the study was embedded posed the greatest challenge and ultimately led to early termination of recruitment.

**Trial registration number** ISRCTN15875290; pre-results

## INTRODUCTION

The threat of the novel COVID-19 and declaration of a pandemic greatly impacted research activity around the world with many studies and trials paused and recruitment ceased unless COVID-19 related or providing potential life preserving interventions.[1] Pandemic responses led to changes in how clinical services were delivered across health systems, with many usual face to face outpatient activities changing quickly to be remotely delivered via telephone or video.[2]

Reports have detailed how clinical services and clinical trials adapted in response to the need to lower the threat posed by COVID-19,[3 4] initially through reduced face-to-face attendance at healthcare institutions and protecting the most clinically vulnerable by promoting minimal activity outside of the home, practising social distancing and discouraging use of public transport.[5] In the UK, people considered to be clinically extremely vulnerable, such as those with haematological cancers, were advised to shield to protect themselves from exposure to infection. As a result, clinical pathways and research protocols required significant adaptations to continue.[6–8] We report the challenges of continuing and adapting a prehabilitation and rehabilitation exercise intervention trial in response to the COVID-19 outbreak and subsequent changes to the clinical pathway into which the trial was integrated. This communication seeks to document changes to our previously published protocol, as well as facilitate study design in the immediate post pandemic era where future interventional trials need to be adaptable in case of unexpected future pandemics or other health emergencies.

## BACKGROUND TO TRIAL

The PERCEPT myeloma prehabilitation study was a pilot, single-centre, randomised controlled trial (RCT) that aimed to investigate the feasibility of a physiotherapist-led exercise intervention, designed to be embedded as an integral part of an established clinical pathway for patients with multiple myeloma undergoing autologous stem cell transplantation (ASCT). Recruitment to the trial commenced at University College Hospital London (UCLH) in June 2019. In summary, all patients referred to the centre for consideration of ASCT for management

of myeloma, who were willing to and clinically able to undertake an exercise training programme and had good command of written and spoken English were eligible and approached. The trial assessments included a range of objective measures of functional capacity (6 min walk test, 30 s timed sit to stand test, hand grip strength test, levels of physical activity) as well as patient-reported outcome measures assessing quality of life (Functional Assessment of Cancer Therapy (FACT) bone marrow transplant and European Organisation for Research and Treatment of Cancer Quality of Life Questionnaire-C30), fatigue (FACT fatigue subscale) and physical activity behaviour (International Physical Activity Questionnaire and exercise self-efficacy scale). Assessments were initially designed to be conducted face to face at established touchpoints within the clinical pathway that usually coincided with patient attendance at transplant clinic appointments. The intervention involved an exercise intervention with incorporated behaviour change techniques, delivered in part through weekly face to face, group-based exercise sessions at a UCLH gym supervised by a physiotherapist and through unsupervised home exercise sessions with telephone support in the post-ASCT rehabilitation phase. Participants were asked to exercise three times per week, at moderate to high intensity, for 30 min of aerobic exercise and complete a programme of resistance exercises. Exercise was individually tailored and progressed by the study physiotherapist. In the prehabilitation phase before ASCT participants were expected to attend weekly supervised gym sessions and exercise independently twice per week. During hospital admission and following discharge from hospital participants were supported to continue three unsupervised exercise sessions per week with weekly telephone guidance from a physiotherapist. The original trial was registered (ISRCTN15875290) and the protocol is described in detail elsewhere.[9]

## TRIAL RECRUITMENT PRIOR TO THE PANDEMIC

Trial recruitment was progressing as planned and between June 2019 and end of February 2020 90 potential participants had been identified and screened for eligibility. Of those, n=36 (40%) consented to take part and were randomised, n=8 (9%) were ineligible and n=46 (51%) declined. The main reasons for declining to take part were travel or distance to take part in the intervention, poor mobility/inability to use public transport and already having too many hospital appointments. Many of those who declined had a favourable opinion of the trial and would have taken part if it was closer to their home. It had been anticipated that target recruitment (n=60–75) would be reached by December 2020.

Face-to-face recruitment was paused in March 2020, due to the COVID-19 pandemic, when all but essential NHS research activity was instructed to cease and research physiotherapists were redeployed to clinical services. In addition, the myeloma ASCT service was also paused due to resource limitations, lack of critical care provision

and increased risk of COVID-19 infection in transplant recipients.

## RESUMING RECRUITMENT AND ADAPTATION TO VIRTUAL INTERVENTION DELIVERY

Between March and June 2020, all participants already enrolled within the trial expressed their wish to continue in the trial while awaiting their delayed ASCT. All participants were placed on holding chemotherapy by their clinical teams. A minor ethics amendment was granted to allow participants to remain in the trial longer than in the original protocol, to continue delivery of the home-based component of the exercise intervention via telephone support and to collect follow-up assessment data via postal delivery of trial questionnaires.

Following resumption of the myeloma ASCT clinical service in June 2020, elements of the clinical pathway on which the trial procedures were designed had changed significantly in response to the ongoing threat of the pandemic. Namely, patients were no longer attending UCLH clinics in person in the lead up to their ASCT and were instead receiving telephone contact for clinic appointments. It was therefore necessary to adapt the trial to accommodate these changes to the clinical pathway.

### Patient and public involvement

Participants who had remained in the study during the first national lock down were consulted on how best to proceed with the trial. These participants were asked to consider their acceptability of receiving study assessments/questionnaires via post, and importantly, how they would feel about only having access to the physiotherapist and exercise intervention via video platform. Overwhelmingly, they supported adapting the trial to virtual delivery and reported that having regular contact with the study physiotherapist was supportive, especially as they were no longer attending their clinical services in person. Some reported that the exercise intervention was something positive to focus on while feeling apprehensive about having their ASCT delayed and remaining limited in their social activities due to shielding. These participants were consulted on the proposed substantial amendment and gave their approval to the changes described in this manuscript.

Following review of emerging literature for adapting pulmonary rehabilitation services to virtual delivery[10–12] and liaising with other cancer rehabilitation services delivering exercise virtually, the trial design was substantially amended.

### Virtual recruitment, consenting and baseline assessments

▶ Identifying potential participants from a list of patients awaiting consideration for ASCT, rather than from weekly clinic lists of expected clinic attendance.

▶ An additional inclusion criterion was added to require patients to have access to and the ability to

use internet-based videoconferencing platform, or a family member to facilitate access.

▶ Participants were sent trial consent forms, self-complete questionnaire measures and a pre-programmed ActivPAL accelerometer by post and a Zoom video call appointment was arranged for approximately 3 days later. Completion of consent forms, baseline questionnaires and fitting of accelerometer discussed and conducted virtually over Zoom appointment.

▶ Addition of the Activities-Specific Balance Confidence Scale[13] to assess risk of falling, used to screen those randomised to the intervention for any necessary actions required to reduce risk of exercising at home as no longer objectively assessed by the physiotherapist face to face. If any participant scored <65% on the scale they would be deemed high risk of falling[14] and their assessment and participation in the group sessions would be adapted to minimise risk.

▶ Objective measure of functional capacity changed to a remotely delivered 1 min timed sit to stand test, which has been demonstrated to induce comparable cardiovascular stress and lower limb muscle fatigue as the 6 min walk test.[15]

### Virtual delivery of the partly supervised exercise intervention

▶ The intervention delivery was changed to be completely online but following the principles of the original protocol in terms of frequency, intensity, time and type of exercise. Each new participant had an initial one to one session to introduce the programme and tailor to their individual abilities, including any symptoms of bone disease or balance deficits. Each participant was then asked to attend a weekly virtual group-based exercise class via Zoom until they were admitted for their ASCT.

### RECRUITMENT TO VIRTUAL TRIAL TO DATE

Recruitment restarted after the first wave of the pandemic in August 2020. Up to October 2020 a further 33 potential participants were identified and screened for eligibility. Of those, 14 (43%) agreed and consented to take part, 6 (18%) were ineligible and 13 (39%) declined. This indicates that uptake to the virtual part of the trial was similar to the original face to face design. Despite fewer people declining the virtual trial, more people were ineligible. The main reason for increased proportion of ineligibility was related to some potential participants already having a date to be admitted for ASCT within the succeeding 2–4 weeks or not having access to the internet.

With the resurgence of COVID-19 in the Autumn of 2020 and as clinical services for ASCT were disrupted again, it was decided to cease recruitment to the trial. The predominant challenge to the adapted trial was variable intervals between timepoints, which differed in length from those detailed in the original study schedule. Further, a small number of participants were withdrawn from the trial as their planned ASCT was deferred or delayed due to clinical risk of proceeding with the ongoing threat of the COVID-19 virus and prioritisation of ASCT admissions based on clinical need in this context. The main aim of the study was to investigate the feasibility of delivering an intervention embedded within an existing clinical pathway. As the pathway on which the trial was designed was significantly affected by the COVID-19 pandemic, which would likely continue to affect the pathway for some time, it was agreed by the study team that little more would be gained by recruiting further. The trial continued in follow-up until all those recruited either reached the final timepoint following ASCT or were withdrawn. The challenges to timely recruitment and delivery of the trial time points as based on the clinical ASCT pathway have implications on study design and will be addressed with the full reporting of the trial which is expected in 2022.

### CONCLUSION

The emergence of the COVID-19 pandemic required significant adaptation for the continuation of this pilot RCT. Our sample population being clinically extremely vulnerable were advised to shield, as well as the requirement to reduce all but clinically necessary face to face contact altered how potential ASCT recipients accessed the clinical setting and consequently the planned trial delivery. We have briefly described how the trial protocol was amended and continued to recruit to and deliver a prehabilitation intervention virtually. Adaptation to virtual delivery was deemed important to participants already enrolled within the trial and appears to have been acceptable for the adapted virtual version of the trial while this was possible.

The original trial design relied heavily on participant attendance at face-to-face clinical contacts as part of the routine ASCT pathway as it existed prepandemic. This was a strength in terms of increasing likely participation in follow-up study assessments and ensuring completeness of data capture as participants would be present in the centre at the planned follow-up time points. However, the assumption that all assessments would be conducted face to face resulted in a reliance on paper-based questionnaires and because study resources did not allow for transformation to electronic data capture, postal follow-up assessment was required. On reflection, the original study would have been somewhat future proofed had electronic questionnaires and intervention materials been incorporated at design and should certainly be a requirement of future work.

As ASCT recipients with myeloma are generally older, with high incidence of disease-related bone disease, deconditioned by induction chemotherapy with possible impairments to mobility and function, adapting the trial to explore feasibility of a virtually delivered physiotherapy intervention was an important opportunity arising from the threat posed by the pandemic. Considering the possible increased risk from not assessing participants face to face, required implementation of a remotely delivered

balance assessment tool, an alternate objective measure of functional capacity suitable for remote assessment and on-line adaptation of intervention delivery to include a virtual group-based exercise class were all important steps. Development of effective, remotely delivered rehabilitation and physical activity programmes are likely to be increasingly valuable to people living with myeloma for whom shielding during clinically vulnerable periods of treatment and recovery, and travel to their specialist treatment centres will be continuing challenges in the post pandemic era.

**Acknowledgements** The authors wish to thank the people living with myeloma who provided input into the original and amended study designs.

**Contributors** OM drafted this manuscript, with input and final approval from KY, AF, GR, ALR, JL, KR and NR. OM, KY, AF and GR designed and gained ethical approval for the original and amended study protocols. ALR, JL, NR and KR provided intellectual input to the study protocols according to their area of expertise. All authors contributed to design of the study, conduct of the study and are accountable for all aspects of the work.

**Funding** This communication relates to independent research supported by the National Institute for Health Research (HEE/ NIHR ICA Programme Clinical Doctoral Research Fellowship, Ms Orla McCourt, ICACDRF-2017-03-067).

**Disclaimer** The views expressed in this publication are those of the author(s) and not necessarily those of the NHS, the National Institute for Health Research or the Department of Health and Social Care.

**Competing interests** None declared.

**Patient consent for publication** Not required.

**Ethics approval** This study involves human participants and was approved by NHS REC London - Camden & Kings Cross reference 19/LO/0204 Participants gave informed consent to participate in the study before taking part.

**Provenance and peer review** Not commissioned; externally peer reviewed.

**ORCID iD**
Orla McCourt http://orcid.org/0000-0001-7572-2540

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
