## [Reviewer comments · BMJ Open]

ARTICLE DETAILS

TITLE (PROVISIONAL)	Adaptation of the PERCEPT myeloma prehabilitation trial to virtual delivery: changes in response to the COVID-19 pandemic
AUTHORS	McCourt, Orla; Fisher, Abigail; Ramdharry, Gita; Roberts, Anna; Land, Joanne; Rabin, Neil; Rowe, Katie; Yong, Kwee

VERSION 1 – REVIEW

REVIEWER	Stout, Nicole West Virginia University Cancer Institute, Hematology/Oncology Cancer Prevention and Control
REVIEW RETURNED	29-Dec-2021

GENERAL COMMENTS	Thank you for providing the detailed adaptations to your clinical trial in this manuscript. While it is unfortunate that your study team has faced these challenges, there is much to be learned from this manuscript through the program adaptations and rationale you offer in this text. A few minor comments: Introduction: - 2nd paragraph: "Reports have detailed..." this sentence describes that there is evidence for the requisite adaptations to clinical trials, however no reference or reports are cited.- 2nd paragraph - last sentence: the language is confusing. "...where future proofing interventional trials to adapt to future pandemic..." it is unclear what this means. Recruitment of Virtual Trial to Date - you note in the final sentence that "...reporting of the trial is expected in early 2021." Is this mistaken and should it be 2022? Or has the report already published?
--

REVIEWER	Chang, Philip Cedars-Sinai Medical Center Samuel Oschin Comprehensive Cancer Institute, Physical Medicine and Rehabilitation
REVIEW RETURNED	11-Jan-2022

GENERAL COMMENTS	This is a very interesting topic that would be relevant to anyone involved in clinical trial design. The novel insight of future-proofing clinical trials is very interesting and has not been sufficiently considered in the literature. The article is mostly well written and well-presented although may benefit from some organizational changes and additions. This article could help medical researchers to make better decisions regarding their protocol planning. Abstract: -Would include in the summary that ultimately the transition to the virtual format was not possible due to the changes in the established
--

	ASCT pathway Strengths and Limitations -It is unclear if this section is needed as this was not a fully completed study Background to Trial -It would be helpful to briefly review the original PERCEPT myeloma protocol in more detail including the inclusion criteria, exact intervention (FITT protocol), the exact objective measures, and the exact PROMs. Resuming Recruitment and Adaptation to Virtual Intervention Delivery -Why was postal delivery of the trial questionnaires chosen opposed to electronic which would reduce time delays and allow better adherence to the protocol timeline? -What was the proposed exact virtual intervention (FITT protocol), exact objective measures, and the exact PROMs -It may be useful to have a graphic or table if possible comparing the original protocol to the virtual protocol and to detail why each change was made regarding the outcome measures used Virtual recruitment, consenting and baseline assessments -The third bullet point beginning with "Trial consent forms..." is difficult to understand and would benefit from rephrasing -In evaluating for fall risk with the one-minute sit to stand was there a threshold at which patients would not be able to participate if their fall risk was too high? Discussion/Conclusion -This article brings up an extremely important point of how we need to be redesigning our trials to future-proof against unexpected turns with the pandemic -It would be very helpful to get the authors' insights on what they think they would have changed about their original protocol if they knew what they know now or even if they would not have proceeded with the study in the first place in hindsight
--	--

VERSION 1 – AUTHOR RESPONSE

Reviewer 1 comments:

Thank you for providing the detailed adaptations to your clinical trial in this manuscript. While it is unfortunate that your study team has faced these challenges, there is much to be learned from this manuscript through the program adaptations and rationale you offer in this text.

Response: Thank you for your supportive comments regarding our manuscript.

A few minor comments:

Introduction:

- 2nd paragraph: "Reports have detailed..." this sentence describes that there is evidence for the requisite adaptations to clinical trials, however no reference or reports are cited.

Response: These citations were omitted by mistake. We have added the citations related to clinical trials and clinical service changes as well as a citation for the relevant policy responses.

- 2nd paragraph - last sentence: the language is confusing. "...where future proofing interventional trials to adapt to future pandemic..." it is unclear what this means.

Response: We have reworded the sentence to "...where future interventional trials need to be adaptable in case of unexpected future pandemics or other health emergencies."

Recruitment of Virtual Trial to Date

- you note in the final sentence that "...reporting of the trial is expected in early 2021." Is this mistaken and should it be 2022? Or has the report already published?

Response: Thank you for highlighting this error. We have updated the year to 2022 and removed 'early', as the trial analysis is still underway.

Reviewer 2 comments:

This is a very interesting topic that would be relevant to anyone involved in clinical trial design. The novel insight of future-proofing clinical trials is very interesting and has not been sufficiently considered in the literature. The article is mostly well written and well-presented although may benefit from some organizational changes and additions. This article could help medical researchers to make better decisions regarding their protocol planning.

Response: Thank you for your positive comments and support for this manuscript.

Abstract:

-Would include in the summary that ultimately the transition to the virtual format was not possible due to the changes in the established ASCT pathway

Response: The final sentence of the abstract summary was edited to- "...however continued changes to the clinical pathway within which the study was embedded posed the greatest challenge and ultimately led to early termination of recruitment."

Strengths and Limitations

-It is unclear if this section is needed as this was not a fully completed study

Response: This section has been removed on your suggestion and the suggestion of the editor.

Background to Trial

-It would be helpful to briefly review the original PERCEPT myeloma protocol in more detail including the inclusion criteria, exact intervention (FITT protocol), the exact objective measures, and the exact PROMs.

Response: Additional summary details of the original study inclusion criteria, intervention, and outcome measures have been added and the citation for the published protocol placed at the end of this section.

"In summary, all patients referred to the centre for consideration of ASCT for management of myeloma, who were willing to and clinically able to undertake an exercise training programme and had good command of written and spoken English were eligible and approached. The trial assessments included a range of objective measures of functional capacity (six minute walk test, 30 second timed sit to stand test, hand grip strength test, levels of physical activity) as well as patient reported outcome measures assessing quality of life (QOL) (Functional Assessment of Cancer Therapy (FACT) bone marrow transplant and European Organisation for Research and Treatment of Cancer QLQ-C30), fatigue (FACT fatigue subscale) and physical activity behaviour (International Physical Activity Questionnaire and exercise self-efficacy scale)."

"The intervention involved an exercise intervention with incorporated behaviour change techniques, delivered in part through weekly face-to-face, group-based exercise sessions at a UCLH gym supervised by a physiotherapist and through unsupervised home exercise sessions with telephone support in the post-ASCT rehabilitation phase. Participants were asked to exercise three times per

week, at moderate to high intensity, for 30 minutes of aerobic exercise and complete a programme of resistance exercises.”

Resuming Recruitment and Adaptation to Virtual Intervention Delivery

-Why was postal delivery of the trial questionnaires chosen opposed to electronic which would reduce time delays and allow better adherence to the protocol timeline?

Response: Unfortunately, study resource and time required to adapt materials to electronic format required us to continue using the original paper instruments and use a postal procedure. We have added a reflection on this limitation to the conclusion section in response to your final comment (see below).

-What was the proposed exact virtual intervention (FITT protocol), exact objective measures, and the exact PROMs

-It may be useful to have a graphic or table if possible comparing the original protocol to the virtual protocol and to detail why each change was made regarding the outcome measures used

Response: A sentence has been added to highlight that the principles of the virtual intervention were the same as the original protocol.

“The intervention delivery was changed to be completely online *but following the principles of the original protocol in terms of frequency, intensity, time and type of exercise.*”

Virtual recruitment, consenting and baseline assessments

-The third bullet point beginning with “Trial consent forms...” is difficult to understand and would benefit from rephrasing

Response: Sentence reordered to clarify

“*Participants were sent trial consent forms, self-complete questionnaire measures and a pre-programmed ActivPAL accelerometer by post and a Zoom video call appointment was arranged for approximately 3 days later.*”

-In evaluating for fall risk with the one-minute sit to stand was there a threshold at which patients would not be able to participate if their fall risk was too high?

Response: The following sentence has been added

“If any participant scored <65% on the scale they would be deemed high risk of falling(Lajoie and Gallagher, 2004) and their assessment and participation in the group sessions would be adapted to minimise risk.”

Discussion/Conclusion

-This article brings up an extremely important point of how we need to be redesigning our trials to future-proof against unexpected turns with the pandemic

-It would be very helpful to get the authors’ insights on what they think they would have changed about their original protocol if they knew what they know now or even if they would not have proceeded with the study in the first place in hindsight

Response: A short paragraph has been added to provide reflection on reliance on paper measures and lack of electronic data capture, that would have allowed the study to adapt in a more efficient way.

“The original trial design relied heavily on participant attendance at face to face clinical contacts as part of the routine ASCT pathway as it existed pre-pandemic. This was a strength in terms of increasing likely participation in follow up study assessments and ensuring completeness of data capture as participants would be present in the centre at the planned follow up timepoints. However, the assumption that all assessments would be conducted face to face resulted in a reliance on paper-based questionnaires and because study resources did not allow for transformation to electronic data capture, postal follow up assessment was required. On reflection, the original study would have been somewhat future proofed had electronic questionnaires and intervention materials been incorporated at design and should certainly be a requirement of future work.”